# An Information-Geometric Distance on the Space of Tasks

## Abstract

We compute a distance between tasks modeled as joint distributions on data and labels. We develop a stochastic process that transports the marginal on the data of the source task to that of the target task, and simultaneously updates the weights of a classifier initialized on the source task to track this evolving data distribution. The distance between two tasks is defined to be the shortest path on the Riemannian manifold of the conditional distribution of labels given data as the weights evolve. We derive connections of this distance with Rademacher complexity-based generalization bounds; distance between tasks computed using our method can be interpreted as the trajectory in weight space that keeps the generalization gap constant as the task distribution changes from the source to the target. Experiments on image classification datasets verify that this task distance helps predict the performance of transfer learning and shows consistency with fine-tuning results. [1]

## 1   Introduction

A part of the success of Deep Learning stems from the fact that deep networks learn features that are discriminative yet flexible. Models pre-trained on a task can be easily transferred to perform well on other tasks. There are also situations when transfer learning does not work. For instance, a pre-trained ImageNet model is a poor representation to transfer to images in radiology (Merkow et al., 2017). It stands to reason that if source and target tasks are "close" to each other then we should expect transfer learning to work well. We lack theoretical tools that define when two learning tasks are close to each other. We also lack algorithmic tools to robustly transfer models on new tasks, for instance, fine-tuning methods require careful design (Li et al., 2020) and it is unclear what one should do if they do not work.

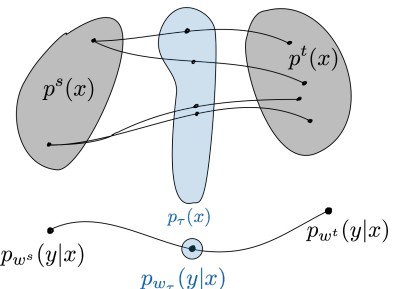

Figure 1: **Coupled transfer of the data and the conditional distribution**. We solve an optimization problem that transports the source data distribution $p^s(x)$ to the target distribution $p^t(x)$ as $\tau \to 1$ while simultaneously updating the model using samples from the interpolated distribution $p_\tau(x)$. Distance between source and target tasks is defined to be the length of the optimal trajectory of the weights under the Fisher Information Metric.

Our approach on these problems is based on the following two ideas: First, most algorithmic methods that devise a distance between tasks in the current literature, do not take into account the hypothesis space of the model. We argue that transferring the representation of a small model with limited capacity from a source task to a target task is difficult because there are fewer redundant features in the model. The distance between the same two tasks may be small if a high-capacity model is being transferred. A definition of task distance or an algorithm for instantiating this definition, therefore needs to take the capacity of the hypothesis class into account.

Second, distance between tasks as measured using techniques in the current literature does not take into account the dynamics of learning; distances often depend on how the transfer was performed. For instance, if one considers the number of epochs of fine-tuning required to reach a certain accuracy, a different strategy may result in a different distance. A sound notion of distance between tasks should not depend on the specific dynamical process used of transfer.

---

[1] See the Appendix for the longer version of the paper.

## 2 Theoretical setup

Consider a dataset $\widehat{p}^s = \{(x_i, y_i) \sim p^s\}_{i=1,\ldots,N_s}$ where $x_i \in X, y_i \in Y$ denote input data and their ground-truth annotations respectively. Data are drawn from a distribution $p^s$ supported on $X \times Y$. Training a parameterized model, say a deep network, involves minimizing the cross-entropy loss $\ell^s(w) := -\frac{1}{N_s} \sum_{i=1}^{N_s} \log p_w(y_i | x_i)$ using a sequence of weight updates $\mathrm{d}w(\tau)/\mathrm{d}\tau = -\widehat{\nabla}\ell^s(w); \; w(0) = w^s$; these are typically implemented using Stochastic Gradient Descent (SGD). The notation $\widehat{\nabla}\ell^s(w)$ indicates a stochastic estimate of the gradient. We will denote the marginal of the joint distributions on the input data as $p^s(x)$ and $p^t(x)$ respectively.

### 2.1 Fisher-Rao metric on the manifold of probability distributions

Consider a manifold $M = \{p_w(z) : w \in \mathbb{R}^p\}$ of probability distributions. Information Geometry (Amari, 2016) studies invariant geometrical structures on such manifolds. For two points $w, w' \in M$, we can use the Kullback-Leibler (KL) divergence $\mathrm{KL}\,[p_w, p_{w'}] = \int \mathrm{d}p_w(z) \log\left(p_w(z)/p_{w'}(z)\right),$ to obtain a Riemannian structure on $M$. Such a structure allows the infinitesimal distance $\mathrm{d}s$ on the manifold to be written as

$$\mathrm{d}s^2 = 2\mathrm{KL}\,[p_w, p_{w+\mathrm{d}w}] = \sum_{i,j=1}^{p} g_{ij}\, \mathrm{d}w_i \mathrm{d}w_j \tag{1}$$

$$g_{ij}(w) = \int \mathrm{d}p_w(z)\, (\partial_{w_i} \log p_w(z))\, (\partial_{w_j} \log p_w(z)) \tag{2}$$

where the Fisher Information Matrix (FIM) $(g_{ij})$ is positive-definite. Given a continuously differentiable curve $\{w(\tau)\}_{\tau \in [0,1]}$ on the manifold $M$ we can compute its length by integrating the infinitesimal distance $|\mathrm{d}s|$ along it. The shortest length curve(geodesic) between two points $w, w' \in M$ induces a metric on $M$ known as the Fisher-Rao distance (Rao, 1945)

$$d_{\mathrm{FR}}(w, w') = \min_{w:\, w(0)=w, w(1)=w'} \int_0^1 \sqrt{\langle \dot{w}(\tau), g(w(\tau))\dot{w}(\tau)\rangle}\mathrm{d}\tau; \tag{3}$$

We will only parametrize the conditional, i.e., we write $p_w^s(x, y) := p^s(x)\, p_w(y|x)$. The marginal on input is not parameterized. This is a simplifying assumption and allows us to decouple the data distribution from the conditional. Notice that we do not need to compute the FIM but given a trajectory of weights $\{w(\tau)\}_{\tau \in [0,1]}$ we can compute its length directly by averaging the square root of the KL-divergence between the conditional distributions of labels given data.

### 2.2 Transporting the data distribution

We would like to modify the empirical input distribution from $\widehat{p}^s(x)$ source to $\widehat{p}^t(x)$ target, that consist of finite samples $N_s$ and $N_t$ respectively. We will use tools from optimal transportation (OT) for this purpose and the optimal coupling is

$$\Gamma^* = \operatorname*{argmin}_{\Gamma \in \Pi(\widehat{p}^s, \widehat{p}^t)} \{\langle \Gamma, C\rangle - \epsilon H(\Gamma)\} \tag{4}$$

where $\Pi(\widehat{p}^s, \widehat{p}^t) = \left\{\Gamma \in \mathbb{R}_+^{N_s \times N_t} : \; \Gamma \mathbb{1}_{N_s} = \widehat{p}^s, \Gamma^\top \mathbb{1}_{N_t} = \widehat{p}^t\right\}$ and $C_{ij} = \|x_i^s - x_j^t\|_2^2$ is the pairwise transportation cost between the source and target data. McCann's interpolation Santambrogio (2015) can be written explicitly as the distribution

$$p_\tau(x, y) = \sum_{i=1}^{N_s} \sum_{j=1}^{N_t} \Gamma_{ij}^* \, \delta_{(1-\tau)x_i + \tau x_j'}(x) \, \delta_{(1-\tau)y_i + \tau y_j'}(y). \tag{5}$$

In practice, we compute the cost $C_{ij}$ using a feature generate (a large neural network trained on some some arbitrary task) and implement the peculiar convex combination of the input data in (5) using Mixup (Zhang et al., 2017).

## 3 Methods

For a coupling matrix $\Gamma$, the interpolated distribution corresponding to the transportation cost is given by (5). Observe that since $\Gamma \in \mathbb{R}^{N_s \times N_t}$, the $(ij)^{\text{th}}$ entry of this matrix indicates the interpolation of source input $x_i^s \in \widehat{p}^s$ with that of target input $x_j^t \in \widehat{p}^t$. The distance between two tasks is now be computed for the two *datasets* $\widehat{p}^s$ and $\widehat{p}^t$ iteratively as follows. Given an initialization $\Gamma^0$ computed using some feature extractor output similarities, we perform the following updates at each iteration.

$$\Gamma^{k+1} = \underset{\Gamma \in \Pi}{\operatorname{argmin}} \left\{ \left\langle \Gamma, C^k \right\rangle - \epsilon H(\Gamma) - \lambda^{-1} \left\langle \Gamma, \Gamma^k \right\rangle \right\}, \tag{6a}$$

$$C_{ij}^k = \int_0^1 \sqrt{2\text{KL}\left[ p_{w_\tau^k}(\cdot|x_{ij}^\tau), p_{w_{\tau+\text{d}\tau}^k}(\cdot|x_{ij}^\tau) \right]}, \tag{6b}$$

$$\frac{\text{d}w_\tau^{k+1}}{\text{d}\tau} = \widehat{\nabla}_w \left\{ \underset{(x,y)\sim p_\tau}{\mathbb{E}} \left[ \log p_{w_\tau^{k+1}}(y|x) \right] \right\}, \tag{6c}$$

$$p_\tau(x,y) = \sum_{i=1}^{N_s} \sum_{j=1}^{N_t} \Gamma_{ij}^k \, \delta_{(1-\tau)x_i^s + \tau x_j^t}(x) \, \delta_{(1-\tau)y_i^s + \tau y_j^t}(y). \tag{6d}$$

At each iteration, the matrix of costs $C_{ij}^k$ is used to store the cost of transporting the input $x_i^s$ to $x_j^t$ along the weight trajectory $\left\{ w_\tau^k \right\}_{\tau \in [0,1]}$ obtained in (6c); all trajectories are initialized at $w^k(0) = w^s$. Observe that the transport cost has the length of the trajectory in weight space (the integral in (3)) incorporated into it. This is our current candidate for the OT cost matrix, similar to one in (4). Given these costs, we can compute the new coupling matrix $\Gamma^{k+1}$ using (6a) which is in turn used in the next iteration to compute the interpolated distribution $p_\tau$ via (6d). Computing the task distance is a non-convex optimization problem and we therefore include a proximal term in (6a) to keep the coupling matrix and trajectory of weights close to the one in the previous step. The task distance computed in (6) is asymmetric.

**Remark 1 (Fisher-Rao distance can be compared across different architectures).** The length of the shortest path between two points on the manifold of distributions $p_w(y|x)$, namely the Fisher-Rao distance, does not depend on the number of parameters of the neural architecture. This enables a desirable property: for the same two tasks, task distance using our approach is numerically comparable across different architectures.

## 4 Experimental evidence

We use the CIFAR-10, CIFAR-100 datasets and thier subsets for our experiments. We show results using an 8-layer convolutional neural network along with a larger wide-residual-network-16-4 (Zagoruyko and Komodakis, 2016). The first baseline is Task2Vec (Achille et al., 2019); the second baseline (fine-tuning) directly computes the length of the trajectory in the weight space, i.e., $\int |\text{d}w|$, the trajectory is truncated when validation accuracy on the target task is 95% of its final validation accuracy; the third baseline (uncoupled transfer) uses a mixture of the source and target data, where the interpolating parameter is sampled from $\text{Beta}(\tau, 1-\tau)$ Length of the trajectory is computed using the FIM metric.

**Transferring between CIFAR-10 and CIFAR-100.** We consider four tasks (i) all vehicles in CIFAR-10, (ii) the remainder, namely six animals in CIFAR-10, (iii) the entire CIFAR-10 dataset and (iv) the entire CIFAR-100 dataset. We show results in Fig. 2 using 4×4 distance matrices where numbers in each cell indicate the distance between the source(row) and the target(column) tasks.

Coupled transfer shows similar trends as fine-tuning, e.g., the tasks animals and vehicles-CIFAR-10 are close to

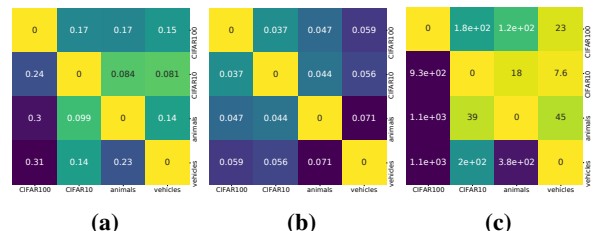

**(a)**     **(b)**     **(c)**

**Figure 2:** Fig. 2a shows distances computed using coupled transfer process, Fig. 2b using Task2Vec while Fig. 2c shows distance using fine-tuning. Numerical values of the distances are not comparable with each other. Coupled transfer distances satisfy certain sanity checks, e.g., transferring to a subset task is easier than transferring from a subset task (CIFAR-10-vehicles/animals).

121 each other while CIFAR-100 is far
122 away. Task distance is asymmetric in Fig. 2a, Fig. 2c. Task2Vec distance estimates in Fig. 2b are
123 qualitatively quite different from these two; the distance matrix is symmetric. Also, while fine-tuning
124 from animals-vehicles is relatively easy, Task2Vec estimates the distance between them to be the
125 largest.

126 **Transferring among subsets of CIFAR-100.** We construct five tasks (herbivores, carnivores,
127 vehicles-1, vehicles-2 and flowers) that are subsets of the CIFAR-100 dataset. Each of these tasks
128 consists of 5 sub-classes. The distance matrices for coupled transfer, Task2Vec and fine-tuning are
129 shown in Fig. 3a, Fig. 3b and Fig. 3c respectively. We also show results using uncoupled transfer
130 in Fig. 3d.

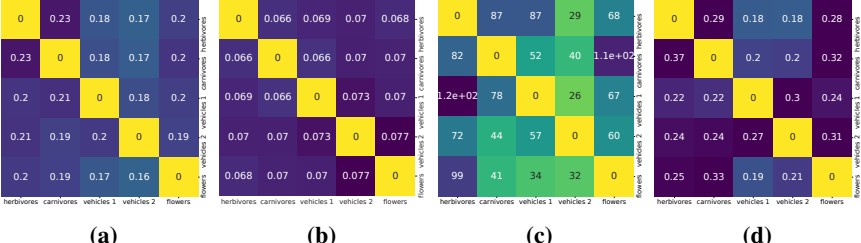

(a) (b) (c) (d)

**Figure 3:** Fig. 3a shows distance for coupled transfer, Fig. 3b shows distance for Task2Vec, Fig. 3c shows
distance for fine-tuning and Fig. 3d shows distance for uncoupled transfer. Numerical values the first and the last
sub-plot can be compared directly. Coupled transfer broadly agrees with fine-tuning except for carnivores-flowers
and herbivores-vehicles-1. For all tasks, uncoupled transfer overestimates the distances compared to Fig. 3a.

131 Coupled transfer estimates that all these sub-
132 sets of CIFAR-100 are roughly equally far away
133 from each other with herbivores-carnivores be-
134 ing the farthest apart while vehicles-1-vehicles-2
135 being closest. This ordering is consistent with
136 the fine-tuning distance although fine-tuning re-
137 sults in an extremely large value for carnivores-
138 flowers and vehicles-1-herbivores. This order-
139 ing is mildly inconsistent with the distances
140 reported by Task2Vec in Fig. 3b the distance
141 for vehicles-1-vehicles-2 is the highest here.
142 Broadly, Task2Vec also results in a distance ma-
143 trix that suggests that all tasks are equally far
144 away from each other. Recall that distances for
145 uncoupled transfer in Fig. 3d can be compared
146 directly to those in Fig. 3a for coupled transfer.

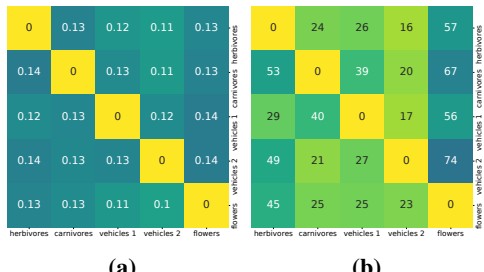

(a) (b)

**Figure 4:** Fig. 4a shows the task distance using coupled
transfer and Fig. 4b show the fine-tuning task distance.
The numbers in Fig. 4a can be directly compared to
those in Fig. 3a. The larger WRN-16-4 model predicts
a smaller task distance for all pairs compared to the
smaller convolutional network in Fig. 3a.

147 Task distances for the former are always larger. This demonstrates the utility of solving a coupled
148 optimization problem in (6) which finds a shorter trajectory on the statistical manifold.

149 **Larger capacity results in smaller task distance.** Task distances for coupled transfer in Fig. 4a are
150 consistent with those for fine-tuning in Fig. 4b. Coupled transfer distances in Fig. 4a are much smaller
151 compared to those in Fig. 3a. This experiment also demonstrates that the information-geometric
152 distance computed by coupled transfer can be compared directly across different architectures; this is
153 not so for most methods in the literature to compute distances between tasks. This gives a constructive
154 strategy for selecting architectures for transfer learning.

## 5 Discussion

156 Our work is an attempt to theoretically understand when transfer is easy and when it is not. An often
157 over-looked idea in large-scale transfer learning is that the dataset need not remain fixed to the target
158 task during transfer. We heavily exploit this idea in the present paper and develop an optimization
159 framework to adapt both the input data distribution and the weights from the source to the target.
160 Although a metric is never unique, this gives legitimacy to our task distance. We compute the *shortest*
161 distance in information space, i.e., the manifold of the conditional distributions. It is remarkable that
162 this concept is closely related to the intuitive idea that a good transfer algorithm is one that keeps the
163 generalization gap small during transfer, in particular at the end on the target task.

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

# An Information-Geometric Distance on the Space of Tasks

**Anonymous Author**
Anonymous Institution

## Abstract

This paper computes a distance between tasks modeled as joint distributions on data and labels. We develop a stochastic process that transports the marginal on the data of the source task to that of the target task, and simultaneously updates the weights of a classifier initialized on the source task to track this evolving data distribution. The distance between two tasks is defined to be the shortest path on the Riemannian manifold of the conditional distribution of labels given data as the weights evolve. We derive connections of this distance with Rademacher complexity-based generalization bounds; distance between tasks computed using our method can be interpreted as the trajectory in weight space that keeps the generalization gap constant as the task distribution changes from the source to the target. Experiments on image classification datasets show that this task distance helps predict the performance of transfer learning: fine-tuning techniques have an easier time transferring to tasks that are close to each other under our distance.

## 1 Introduction

A part of the success of Deep Learning stems from the fact that deep networks learn features that are discriminative yet flexible. Models pre-trained on a task can be easily adapted to perform well on other tasks. The transfer learning literature forms an umbrella for such adaptation techniques. Transfer learning indeed works very well, see for instance Mahajan et al. (2018); Dhillon et al. (2020); Kolesnikov et al. (2019); Joulin et al. (2016) for image classification or Devlin et al. (2018) for language modeling, to name a few of the many large-scale demonstrations. There are however also situations when transfer learning does not work well. For instance, a pre-trained ImageNet model is a poor representation to transfer to images in radiology (Merkow et al., 2017).

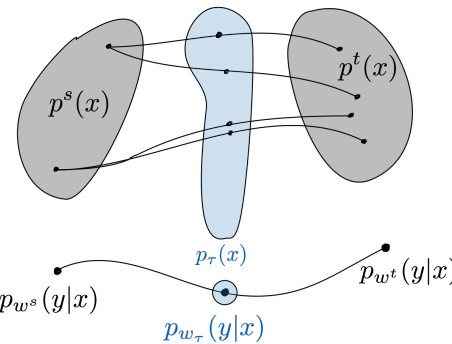

**Figure 1: Coupled transfer of the data and the conditional distribution**. We solve an optimization problem that transports the source data distribution $p^s(x)$ to the target distribution $p^t(x)$ as $\tau \to 1$ while simultaneously updating the model using samples from the interpolated distribution $p_\tau(x)$. This modifies the conditional distribution $p_{w^s}(y|x)$ on the source task to the corresponding distribution on the target task $p_{w^t}(y|x)$. Distance between source and target tasks is defined to be the length of the optimal trajectory of the weights under the Fisher Information Metric.

It stands to reason that if source and target tasks are "close" to each other then we should expect transfer learning to work well; it may be difficult to transfer across tasks that are "far away". We lack theoretical tools that define when two learning tasks are close to each other; while there are numerous candidates in the current literature a unified understanding of these domain-specific methods remains elusive. We also lack algorithmic tools to robustly transfer models on new tasks, for instance, fine-tuning methods require careful design (Li et al., 2020) and it is unclear what one should do if they do not work well. Our approach on this problem is based on the following two ideas.

**Main ideas** First, most algorithmic methods that devise a distance between tasks in the current literature, do not take into account the hypothesis space of the model. We argue that transferring the representation of a small model with limited capacity from a source task to a target task is difficult because there are fewer redundant features in the model. The distance between the same two tasks may be small if a high-capacity model is being transferred; this is especially pertinent when transferring deep networks. A definition of task distance or an algorithm for instantiating

this definition, therefore needs to take the capacity of the hypothesis class into account.

Second, distance between tasks as measured using techniques in the current literature does not take into account the dynamics of learning; distances often depend on how the transfer was performed. For instance, if one considers the number of epochs of fine-tuning required to reach a certain accuracy, a different strategy may result in a different distance. A sound notion of distance between tasks should not depend on the specific dynamical process used of transfer.

**Summary of contributions**   Given data $x$ and labels $y$, we model the source and target tasks as joint distributions $p^s(x, y)$ and $p^t(x, y)$ respectively where

$$p^s(x, y) = p^s_{w^s}(y|x)\, p^s(x);$$

here $w^s$ are the parameters (weights) of a classifier on the source task. The target task is decomposed analogously. We define the distance between finite-sample *datasets* $\widehat{p}^s = \{(x_i, y_i) \sim p^s\}_{i=1}^{N_s}$ drawn from tasks (with $\widehat{p}^t$ defined analogously) as the solution of the optimization problem

$$\min_{\Gamma \in \Pi} \int_0^1 \mathbb{E}_{(x,y) \sim p_\tau(x,y)} \left[ \sqrt{2\mathrm{KL}\left(p_{w_\tau}(\cdot|x),\ p_{w_{\tau+\mathrm{d}\tau}}(\cdot|x)\right)} \right] \tag{1}$$
$$\text{subject to } \frac{\mathrm{d}w_\tau}{\mathrm{d}\tau} = \widehat{\nabla}_w \left\{ \mathbb{E}_{(x,y) \sim p_\tau} \left[ \log p_{w_\tau}(y|x) \right] \right\};$$

where

$$p_\tau(x, y) = \sum_{i=1}^{N_s} \sum_{j=1}^{N_t} \Gamma_{ij}\, \delta_{\left\{(1-\tau)x_i^s + \tau x_j^t\right\}}\, \delta_{\left\{(1-\tau)y_i^s + \tau y_j^t\right\}},$$
$$\Pi = \left\{ \Gamma \in \mathbb{R}_+^{N_s \times N_t} : \Gamma \mathbb{1} = \widehat{p}^s,\ \Gamma^\top \mathbb{1} = \widehat{p}^t \right\}. \tag{2}$$

The square root of the KL-divergence measures the infinitesimal distance traveled along a geodesic on the manifold of probability distributions $p_{w_\tau}(y|x)$. Distance as defined in (1) and (2) is the path length on this manifold. Weights $w(\tau)$ evolve from their initial value $w^s$ using stochastic gradient descent-based updates in (1) while the interpolated data distributed $p_\tau$ to which weights are being fitted to evolves simultaneously from its initial value $p^s$ to its final value $p^t$ using the linear coupling matrix $\Gamma$.

The distance in (1) is the Fisher-Rao distance on the manifold of distributions $p_w(y|x)$. We use this to draw a link to Rademacher complexity-based generalization bounds which gives an intuitive understanding of the trajectory of weights computed in (1). We show that our approach modifies the task's data distribution and weights so as to to minimize the *average generalization gap* along the trajectory that joins the weights on the source task $w^s$ to their terminal value on the target task $w(1)$.

We devise an algorithmic procedure to solve the optimization problem in (1) and (2) for image classification tasks that are subsets of CIFAR-10 and CIFAR-100

datasets (Krizhevsky and Hinton, 2009). We show in Sec. 4 that the coupled transfer process estimates distances that are more consistent with the difficulty of fine-tuning, as compared to baselines.

## 2   Theoretical setup

Consider a dataset $\widehat{p}^s = \{(x_i, y_i) \sim p^s\}_{i=1,\ldots,N_s}$ where $x_i \in X, y_i \in Y$ denote input data and their ground-truth annotations respectively. Data are drawn from a distribution $p^s$ supported on $X \times Y$. Training a parameterized model, say a deep network, involves minimizing the cross-entropy loss $\ell^s(w) := -\frac{1}{N_s} \sum_{i=1}^{N_s} \log p_w(y_i|x_i)$ using a sequence of weight updates

$$\mathrm{d}w(\tau)/\mathrm{d}\tau = -\widehat{\nabla}\ell^s(w);\ w(0) = w^s; \tag{3}$$

these are typically implemented using Stochastic Gradient Descent (SGD). The notation $\widehat{\nabla}\ell^s(w)$ indicates a stochastic estimate of the gradient using a subset of the data.

We will denote the marginal of the joint distributions on the input data as $p^s(x)$ and $p^t(x)$ respectively.

### 2.1   Low-capacity models are difficult to transfer

The Information Bottleneck (IB) Principle (Tishby et al., 2000) abstracts a parametric classifier as a Markov chain $x \to z \to y$ where $z$ is called the representation. The main idea behind IB is to learn sufficient and minimal representations that discard information in the data that are not relevant to predicting labels (Shwartz-Ziv and Tishby, 2017). Gao and Chaudhari (2020) build upon this idea to design a Lagrangian to study transfer learning

$$F_{p^s}(\lambda, \gamma) = \min_{e(z|x),d(x|z),c(y|z)} \{R + \lambda D + \gamma C\}. \tag{4}$$

Here $e(z|x)$ is an encoder that constructs the representation $z$, $c(y|z)$ predicts the labels and $d(x|z)$ is the decoder measures the trade-off between redundant features that give a lossless representation of the data and discriminative features for the classifier. The quantities $R, D$ and $C$ are the rate of the encoder, distortion of the decoder and the classification loss respectively (see **??** for more details).

The situation with a low-capacity model (encoder and classifier) can be modeled as follows. If source and target task are similar then transfer is easy because features of the encoder trained on source are also good for the target. If the tasks are dissimilar, observe that KKT conditions give $\mathrm{d}F_{p^s} = -\mathrm{d}R - \lambda\mathrm{d}D - \gamma\mathrm{d}C$. A low-capacity model needs a large $\gamma$ to achieve the same classification loss $C$ which according to the KKT condition leads to larger value of the $D$ (for fixed $\lambda$). This indicates that the model does not learn redundant features that may be potentially useful on a dissimilar target task. This is not the case for a model with large capacity where (4) learns both redundant and discriminative features, without being forced to make a trade-off between them.

## 2.2 Fisher-Rao metric on the manifold of probability distributions

Consider a manifold $M = \{p_w(z) : w \in \mathbb{R}^p\}$ of probability distributions. Information Geometry (Amari, 2016) studies invariant geometrical structures on such manifolds. For two points $w, w' \in M$, we can use the Kullback-Leibler (KL) divergence KL $[p_w, p_{w'}] = \int dp_w(z) \log (p_w(z)/p_{w'}(z))$, to obtain a Riemannian structure on $M$. Such a structure allows the infinitesimal distance $ds$ on the manifold to be written as

$$ds^2 = 2\text{KL}\,[p_w, p_{w+dw}] = \sum_{i,j=1}^{p} g_{ij}\,dw_i dw_j \quad (5)$$

where the Fisher Information Matrix (FIM) $(g_{ij})$ with each

$$g_{ij}(w) = \int dp_w(z)\,(\partial_{w_i} \log p_w(z))\,(\partial_{w_j} \log p_w(z)) \quad (6)$$

is positive-definite. The weights $w$ play the role of a coordinate system for computing the distance. The FIM is the Hessian of the KL-divergence; we may think of the FIM as quantifying the amount of information present in the model about the data it was trained on. The FIM is the unique metric on $M$ (up to scaling) that is preserved under diffeomorphisms (Bauer et al., 2016). This property motivated Liang et al. (2019) to define a geometric notion of model complexity called the Fisher-Rao norm as

$$\|w\|_{\text{fr}}^2 = \langle w, g\, w \rangle. \quad (7)$$

This will be discussed further in Sec. 3.3.

Given a continuously differentiable curve $\{w(\tau)\}_{\tau \in [0,1]}$ on the manifold $M$ we can compute its length by integrating the infinitesimal distance $|ds|$ along it. The shortest length curve between two points $w, w' \in M$ induces a metric on $M$ known as the Fisher-Rao distance (Rao, 1945)

$$d_{\text{FR}}(w, w') = \min_{\substack{w:\ w(0)=w \\ w(1)=w'}} \int_0^1 \sqrt{\langle \dot{w}(\tau), g(w(\tau))\dot{w}(\tau) \rangle} d\tau; \quad (8)$$

Let us note that shortest paths on a Riemannian manifold are geodesics, i.e., they are locally "straight lines".

**Assumption 1 (Fisher-Rao distance is computed only for the conditional distribution).** Although we are interested in the manifold of joint distributions we will only parametrize the conditional, i.e., we write

$$p_w^s(x, y) := p^s(x)\,p_w(y|x). \quad (9)$$

The marginal on input is not parameterized. This is a simplifying assumption and allows us to decouple the data distribution from the conditional; we can tackle the former using techniques in optimal transport and the latter using techniques in information geometry. Parameterizing the joint distribution directly and using a unified approach to compute the task distance is possible but will require generative modeling of $p^s(x)$ which is computationally challenging.

Under Assumption 1, the FIM in (6) can be written as

$$g_{ij}(w) = \mathbb{E}_{\substack{x \sim p^s(x),\ y \sim p_w(y|x)}} \left[ \partial_{w_i} \log p_w(y|x)\ \partial_{w_j} \log p_w(y|x) \right].$$

The FIM is difficult to compute for large models and approximations often work poorly (Kunstner et al., 2019). Notice that we do not need to compute the FIM but only need to compute the distance $|ds|$. Given a trajectory of weights $\{w(\tau)\}_{\tau \in [0,1]}$ we can compute its length directly by averaging the square root of the KL-divergence between the conditional distributions of labels given data.

## 2.3 Transporting the data distribution

We next focus on the marginals on the input data. We would like to modify the input distribution from $p^s(x)$ to $p^t(x)$ during transfer. We will use tools from optimal transportation (OT) for this purpose; see Santambrogio (2015); Peyré and Cuturi (2019) for an elaborate introduction to OT.

Let $\Pi(p^s, p^t)$ be the set of joint distributions with first marginal equal to $p^s(x)$ and second marginal $p^t(x')$. The Kantorovich relaxation of the OT problem solves for

$$W_2^2(p^s, p^t) = \inf \left\{ \int \|x - x'\|^2\, d\gamma : \gamma \in \Pi(p^s, p^t) \right\}$$

to compute the best joint coupling $\gamma^* \in \Pi$. The solution of this problem is the Wasserstein metric $W_2(p^s, p^t)$ between the two distributions. In this paper, we are interested, not in the Wasserstein metric, but the transport trajectory that the optimal coupling $\gamma^*$ entails. This is the subject of displacement interpolation (McCann, 1997): it turns out that the geodesic that joins $p^s$ and $p^t$ is a locally distance minimizing curve in the $W_2$ metric. If $p_\tau$ is the distribution at an intermediate step $\tau \in [0, 1]$, we have

$$W_2(p^s, p_\tau) = \tau W_2(p^s, p^t).$$

The path that the optimal coupling $\gamma^*$ takes is therefore a *constant-speed* geodesic.

We are interested in instantiating this idea for source and target input *datasets* (we denote these by $\widehat{p}^s(x)$ and $\widehat{p}^t(x)$) that consist of finite samples $N_s$ and $N_t$ respectively. The development is more convenient in this case and the set of transport plans is a (convex) polytope

$$\Pi(\widehat{p}^s, \widehat{p}^t) = \left\{ \Gamma \in \mathbb{R}_+^{N_s \times N_t} : \Gamma \mathbb{1}_{N_s} = \widehat{p}^s, \Gamma^\top \mathbb{1}_{N_t} = \widehat{p}^t \right\} \quad (10)$$

and the optimal coupling is given by

$$\Gamma^* = \operatorname*{argmin}_{\Gamma \in \Pi(\widehat{p}^s, \widehat{p}^t)} \{\langle \Gamma, C \rangle - \epsilon H(\Gamma)\} \quad (11)$$

where $C_{ij} = \|x_i - x_j'\|_2^2$ is the matrix of pairwise distances between the source and target data. The inner product in the first term measures the total cost $\sum_{ij} \Gamma_{ij} C_{ij}$ incurred for the transport and minimizing it directly is typically done

using interior point methods. This can be accelerated using an entropic penalty $H(\Gamma) = -\sum_{ij} \Gamma_{ij} \log \Gamma_{ij}$ popularized by Cuturi (2013). McCann's interpolation for the finite-dimensional case with the quadratic loss $C_{ij}$ can be written explicitly as the distribution

$$p_\tau(x) = \sum_{i=1}^{N_s} \sum_{j=1}^{N_t} \Gamma_{ij}^* \, \delta_{(1-\tau)x_i + \tau x_j'}(x). \qquad (12)$$

Notice that this is a sum of Dirac-delta distributions supported at interpolated *input data* $x = (1-\tau)x_i + \tau x_j'$. We can also create pseudo labels for samples from $p_\tau$ by a linear interpolation of the one-hot encoding of their respective labels to get

$$p_\tau(x,y) = \sum_{i=1}^{N_s} \sum_{j=1}^{N_t} \Gamma_{ij}^* \, \delta_{(1-\tau)x_i + \tau x_j'}(x) \, \delta_{(1-\tau)y_i + \tau y_j'}(y). \qquad (13)$$

**Modifications to the interpolated distribution** We next make two practically motivated modifications to the interpolated distribution $p_\tau(x,y)$.

First, the quadratic distance $C_{ij} = \|x_i - x_j'\|^2$ is not a reasonable notion of visual/text data that have strong local correlations. It is therefore beneficial to compute $C_{ij}$ using a feature extractor, say a large neural network $\varphi$, that is trained on some generic task

$$C_{ij} := \|\varphi(x_i) - \varphi(x_j')\|_2^2. \qquad (14)$$

This gives us a good coupling matrix $\Gamma$ in practice because the feature space $\varphi(x)$ is much more Euclidean-like than the original input space; similar ideas are often employed in the metric learning literature (Snell et al., 2017; Hu et al., 2015; Qi et al., 2018).

Second, the peculiar form of the interpolating distribution in (13) that consists of convex combinations of inputs and labels is the result of the quadratic cost. Samples from such an interpolated distribution will have visual artifacts for image-based data. In practice, we treat the time $\tau$ as parameter of a Beta-distribution $\text{Beta}(\tau, 1-\tau)$. Thereby samples from $p_\tau$ are similar to those created in Mixup regularization Zhang et al. (2017); a fraction $\tau$ of the samples are similar to those from $p^s$ and the remainder are similar to those from $p^t$. Note that which data is used to form the Mixup combinations is still governed by the coupling matrix $\Gamma^*$. We use this trick for both inputs and labels in our experiments.

## 3 Methods

We now combine the development of Sec. 2.2–2.3 to transport the marginal on the data and modify the weights on the statistical manifold. This section also discusses techniques to efficiently implement the approach and make it scalable to large deep networks. Sec. 3.3 discusses an alternate perspective on this coupled transfer process using the connection between Rademacher complexity and the Fisher-Rao norm.

### 3.1 Interpolating tasks using a mixture distribution

Interpolating the source and target tasks using a mixture distribution is a simple way to demonstrate the main idea of our approach. For $\tau \in [0, 1]$, consider

$$p_\tau(x,y) = (1-\tau)p^s(x,y) + \tau p^t(x,y). \qquad (15)$$

This amounts to, on average, $1-\tau$ fraction of samples from $\widehat{p}^s$ and the rest from $\widehat{p}^t$. At time instant $\tau$, weights of the classifier are updated using SGD to fit samples from $p_\tau$. We write this as

$$dw_\tau/d\tau = \widehat{\nabla}_w \underset{(x,y)\sim p_\tau}{\mathbb{E}} \left[\log p_{w_\tau}(y|x)\right]; \ w(0) = w^s \qquad (16)$$

Weights $w_\tau$ can be thought of as fitted to the task $p_\tau$ for every $\tau$, in particular for $\tau = 1$, the weights $w(1)$ is fitted to $p^t$. We can now integrate the length of the trajectory using (8) to compute the distance between tasks.

Changes in the data distribution and updates to the weights are not synchronized in this approach. For instance, changes in the data may be unfavorable to the current weights and this forces a different trajectory in the weight space as the weights struggle to track $p_\tau$. If changes in data were synchronized with those in weights, the weight trajectory would be different *and necessarily shorter* because the KL-divergence in (5) is large if the conditional distribution changes quickly; our experimental results also corroborate this.

### 3.2 Modifying the task and weights simultaneously

We now reintroduce the transport process for the data distribution. For a coupling matrix $\Gamma$, the interpolated distribution corresponding to the squared Euclidean cost in OT is given by (13). Observe that since $\Gamma \in \mathbb{R}^{N_s \times N_t}$, the $(ij)^{\text{th}}$ entry of this matrix indicates the interpolation of source input $x_i^s \in \widehat{p}^s$ with that of target input $x_j^t \in \widehat{p}^t$. The distance between two tasks as defined in (1) and (2) can now be computed for the two *datasets* $\widehat{p}^s$ and $\widehat{p}^t$ iteratively as follows. Given an initialization $\Gamma^0$ computed using a feature extractor in (14), we perform the following updates at each iteration.

$$\Gamma^{k+1} = \underset{\Gamma \in \Pi}{\arg\min} \left\{ \langle \Gamma, C^k \rangle - \epsilon H(\Gamma) - \lambda^{-1} \langle \Gamma, \Gamma^k \rangle \right\}, \qquad (17a)$$

$$C_{ij}^k = \int_0^1 \sqrt{2\text{KL}\left[p_{w_\tau^k}(\cdot|x_{ij}^\tau), p_{w_{\tau+d\tau}^k}(\cdot|x_{ij}^\tau)\right]}, \qquad (17b)$$

$$\frac{dw_\tau^{k+1}}{d\tau} = \widehat{\nabla}_w \left\{ \underset{(x,y)\sim p_\tau}{\mathbb{E}} \left[\log p_{w_\tau^{k+1}}(y|x)\right] \right\}, \qquad (17c)$$

$$p_\tau(x,y) = \sum_{i=1}^{N_s} \sum_{j=1}^{N_t} \Gamma_{ij}^k \, \delta_{(1-\tau)x_i^s + \tau x_j^t}(x) \, \delta_{(1-\tau)y_i^s + \tau y_j^t}(y). \qquad (17d)$$

At each iteration, the matrix of costs $C_{ij}^k$ is used to store the cost of transporting the input $x_i^s$ to $x_j^t$ along the weight

trajectory $\{w_\tau^k\}_{\tau \in [0,1]}$ obtained in (17c); all trajectories are initialized at $w^k(0) = w^s$. Observe that the transport cost has the length of the trajectory in weight space (the integral in (8)) incorporated into it. This is our current candidate for the OT cost matrix, similar to one in (11). Given these costs, we can compute the new coupling matrix $\Gamma^{k+1}$ using (17a) which is in turn used in the next iteration to compute the interpolated distribution $p_\tau$ via (17d). Computing the task distance is a non-convex optimization problem and we therefore include a proximal term in (17a) to keep the coupling matrix close to the one in the previous step $\Gamma^k$. This has the added effect of keeping the entire trajectory of weights $\{w_\tau^{k+1}\}_{\tau \in [0,1]}$ close to the trajectory in the previous iteration. Proximal point iteration (Bauschke and Combettes, 2017) is insensitive to the step-size $\lambda$ and it is therefore beneficial to employ it in these updates.

Let us note that the task distance computed in (17) is asymmetric, the length of the trajectory for transferring from $\widehat{p}^s$ to $\widehat{p}^t$ is different from the one that transfers from $\widehat{p}^t$ to $\widehat{p}^s$.

**Remark 2 (Fisher-Rao distance can be compared across different architectures).** The length of the shortest path between two points on the manifold of distributions $p_w(y|x)$, namely the Fisher-Rao distance, does not depend on the embedding dimension of the manifold $M$. More specifically, the distance between tasks as computed by this length does not depend on the number of parameters of the neural architecture, it only depends upon the capacity to fit the conditional distribution $p_w(y|x)$. This enables a desirable property: for the same two tasks, task distance using our approach is numerically comparable across different architectures.

**Remark 3 (Scaling up the computation).** The formulation in (17) updates $\Gamma \in \mathbb{R}^{N_s \times N_t}$ and $w_\tau \in \mathbb{R}^p$. Executing the updates, even for large deep networks and standard datasets is easy for the weights. The coupling matrix $\Gamma$ has a large number of entries and it is therefore challenging. Some common approaches to handling large-scale OT problems are hierarchical methods (Lee et al., 2019), and greedy computation (Carlier et al., 2010). In practice, we initialize (17a) with a block-diagonal approximation of the coupling matrix using (14) as the costs and perform mini-batch updates on the non-zero entries of $\Gamma$. At each iteration, we sample from the interpolated distribution (17d) using only entries of $\Gamma^k$ that are a part of the mini-batch. Experiments in Sec. 4 show that the weight trajectory converges under such mini-batch updates of $\Gamma^k$.

### 3.3 An alternative perspective via Rademacher complexity

We next study how the trajectory given by the formulation of (17) looks under the lens of learning theory. We show that we can interpret the solution of coupled transfer as a trajectory that minimizes the integral of the generalization gap as the task and the weights are modified. This gives us an intuitive understanding of what qualifies as good transfer; indeed weight trajectories that do not lead to degradation of the generalization gap result in weights on the target task $w^t$ that also generalize well.

We introduce a few quantities before giving the main result. We consider binary classification in this section for clarity. Given a $w \in A$, we define the empirical Rademacher complexity (Bartlett and Mendelson, 2001) as

$$\widehat{\mathcal{R}}_N(A) = \mathbb{E}_\sigma \left[ \sup_{w \in A} \frac{1}{N} \sum_{i=1}^N \epsilon^i \ell(w; x^i, y^i) \right], \qquad (18)$$

where $\sigma^i$ are independent and uniformly distributed on $\{-1, 1\}$ and $\ell(w; x^i, y^i)$ is the loss on the $i^{\text{th}}$ datum of a dataset $\widehat{p}$ with $N$ samples. We will choose the set $A$ to be the $r$-ball in the Fisher-Rao norm

$$A := \{w : \|w\|_{\text{fr}} \le r\},$$

and write the corresponding complexity as $\hat{\mathcal{R}}_N(r)$. The Rademacher complexity is the expectation of the empirical complexity over draws of different datasets

$$\mathcal{R}_N(r) = \mathbb{E}_{\widehat{p} \sim p} \left[ \widehat{\mathcal{R}}_N(r) \right].$$

The classical Rademacher complexity-based generalization bound characterizes the ability of binary classifiers $h$ in a hypothesis class $h \in \mathcal{H}$ to fit random noise. We have that for all $h \in \mathcal{H}$ the absolute value of the generalization error and the training error is upper bounded by

$$\mathcal{R}_{2N}(\mathcal{H}) + 2\sqrt{\frac{\log(1/\delta)}{N}} \qquad (19)$$

with probability at least $1 - \delta$. We build upon this to obtain the following theorem.

**Theorem 4.** Given a trajectory of the weights $\{w_\tau\}_{\tau \in [0,1]}$ and a sequence $0 \le \tau_1 < \tau_2 < ... < \tau_K \le 1$, then for all $\epsilon > \frac{2}{K} \sum_{k=1}^K \mathcal{R}_N(\|w_{t_k}\|_{\text{fr}})$, the probability that

$$\frac{1}{K} \sum_{k=1}^K \left( \mathbb{E}_{(x,y) \sim p_{\tau_k}} [\ell(\omega_{\tau_k}, x, y)] - \frac{1}{N} \sum_{(x,y) \sim \hat{p}_{\tau_k}} \ell(\omega_{\tau_k}, x, y) \right)$$

is greater than $\epsilon$ is upper bounded by

$$\exp\left\{ -\frac{2K}{M^2} \left( \epsilon - \frac{2}{K} \sum_{k=1}^K \mathcal{R}_N(\|w_{t_k}\|_{\text{fr}}) \right)^2 \right\}. \qquad (20)$$

The proof is provided in **??**. In other words, ensuring that the generalization gap of the model is small during transfer can be achieved by ensuring that the Rademacher complexity $\mathcal{R}_N(\|w_\tau\|_{\text{fr}})$ is small at all times during transfer.

**Specializing the result for multi-layer linear models.** Characterizing the Rademacher complexity of the $r$-ball in Fisher-Rao norm is difficult. However, for multi-layer linear classifiers with input domain $X \subset \mathbb{R}^p$, Liang et al. (2019) showed that

$$\mathcal{R}_N(r) \leq r\sqrt{\frac{p}{N}}$$

assuming the data covariance matrix $\mathbb{E}_{x \sim p}[x^\top x]$ is full-rank. For such a model, minimizing the integral of the Rademacher complexity is achieved by minimizing

$$\int_0^1 \sqrt{\langle w_\tau, g(w_\tau)w_\tau \rangle} \, \mathrm{d}\tau; \qquad (21)$$

which is an upper-bound on the Fisher-Rao distance of the trajectory between $p_{w^s}$ and $p_{w^t}$. The optimization problem in (17) finds the latter and we have thus obtained a close connection between the coupled transfer process and the intuitive idea that as the model is transferred, keeping the generalization gap small at all instants of the trajectory would lead to a good generalization gap on the target dataset.

As the authors in Liang et al. (2019) discuss further, the Fisher-Rao norm ball is an envelope of popular norms such as spectral norm (Bartlett et al., 2017), group norm (Neyshabur et al., 2015b) and path norm (Neyshabur et al., 2015a) introduced to characterize the complexity of deep neural networks. The Rademacher complexity using these other norms can therefore be upper-bounded in terms of the Fisher-Rao norm, which leads to a similar conclusion for non-linear models.

# 4 Experimental evidence

This section discusses experiments on image classification datasets which demonstrate that the distance computed using our methods is consistent with the difficulty of fine-tuning from the source dataset to the target dataset. We compare and contrast our distance estimtes with existing methods, discuss how the optimization problem in (17) converges to its solution across iterations and show that larger models are easier to transfer between tasks.

## 4.1 Setup

We use the CIFAR-10, CIFAR-100 datasets for our experiments. Source and target tasks consist of subsets of these datasets, each task with one or more of the original classes inside it. We show results using an 8-layer convolutional neural network with ReLU nonlinearities, dropout, batch-normalization with a final fully-connected layer along with a larger wide-residual-network (WRN-16-4, (Zagoruyko and Komodakis, 2016)). More details of pre-processing, architecture and training procedure are provided in **??**.

**Baselines** The first baseline is Task2Vec (Achille et al., 2019a) which embeds tasks using the diagonal of the FIM

of a model trained on them individually; cosine distance between these vectors is defined as the distance. We compute the robust approximation of the FIM via Monte Carlo updates as done by the original authors.

The second baseline (fine-tuning) directly computes the length of the trajectory in the weight space, i.e., $\int |\mathrm{d}w|$. The trajectory is truncated when validation accuracy on the target task is 95% of its final validation accuracy. Note that no adaptation of input data is performed and the model directly takes SGD updates on the target task after being pre-trained on the source task. The learning rate for each model was tuned across all datasets to ensure that the validation accuracy on the target dataset is good and fixed thenceforth for all experiments. The number of epochs required for fine-tuning is a popular way to measure distance between tasks (Kornblith et al., 2019).

The third baseline (uncoupled transfer) uses a mixture of the source and target data, where the interpolating parameter is sampled from $\mathrm{Beta}(\tau, 1 - \tau)$ (see Sec. 3.1) in order to be consistent with the way we implement coupled transfer and avoid visual artifacts in the input data. Length of the trajectory is computed using the FIM metric in this case, which enables direct comparison of the task distances for coupled and uncoupled transfer.

## 4.2 Transferring between CIFAR-10 and CIFAR-100

We consider four tasks (i) all vehicles (airplane, automobile, ship, truck) in CIFAR-10, (ii) the remainder, namely six animals in CIFAR-10, (iii) the entire CIFAR-10 dataset and (iv) the entire CIFAR-100 dataset. We show results in Fig. 2 using 4×4 distance matrices where numbers in each cell indicate the distance between the source task (row) and the target task (column).

Coupled transfer shows similar trends as fine-tuning, e.g., the tasks animals-CIFAR-10 or vehicles-CIFAR-10 are close to each other while CIFAR-100 is far away from all tasks (it is closer to CIFAR-10 than others). Task distance is asymmetric in Fig. 2a, Fig. 2c. Distance from CIFAR-10-animals is smaller than animals-CIFAR-10; this is expected because animals is a subset of CIFAR-10. Task2Vec distance estimates in Fig. 2b are qualitatively quite different from these two; the distance matrix is symmetric. Also, while fine-tuning from animals-vehicles is relatively easy, Task2Vec estimates the distance between them to be the largest.

This experiment also shows that our approach can scale to medium-scale datasets and can handle situations when the source and target task have different number of classes.

## 4.3 Transferring among subsets of CIFAR-100

We construct five tasks (herbivores, carnivores, vehicles-1, vehicles-2 and flowers) that are subsets of the CIFAR-100 dataset. Each of these tasks consists of 5 sub-classes. The distance matrices for coupled transfer, Task2Vec and fine-tuning are shown in Fig. 3a, Fig. 3b and Fig. 3c respectively.

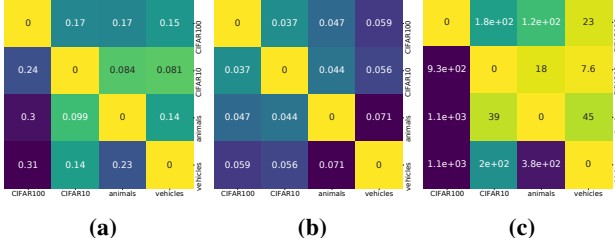

**(a)**        **(b)**        **(c)**

**Figure 2:** Fig. 2a shows distances (numbers in the cell) computed using our coupled transfer process, Fig. 2b shows distances estimated using Task2Vec while Fig. 2c shows the distance estimating using fine-tuning. The numerical values of the distances in this figure are not comparable with each other. Coupled transfer distances satisfy certain sanity checks, e.g., transferring to a subset task is easier than transferring from a subset task (CIFAR-10-vehicles/animals).

We also show results using uncoupled transfer in Fig. 3d.

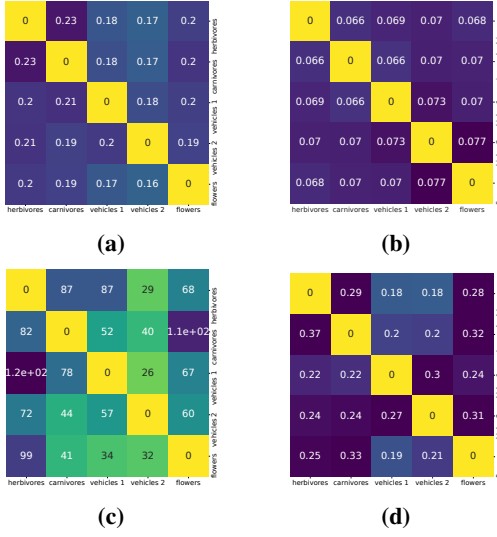

**Figure 3:** Fig. 3a shows the distance for coupled transfer, Fig. 3b shows the distance for Task2Vec, Fig. 3c shows the distance for fine-tuning and Fig. 3d shows the distance for uncoupled transfer. Numerical values the first and the last sub-plot can be compared directly. Coupled transfer broadly agrees with fine-tuning except for carnivores-flowers and herbivores-vehicles-1. For all tasks, uncoupled transfer overestimates the distances compared to Fig. 3a.

Coupled transfer estimates that all these subsets of CIFAR-100 are roughly equally far away from each other with herbivores-carnivores being the farthest apart while vehicles-1-vehicles-2 being closest. This ordering is consistent with the fine-tuning distance although fine-tuning results in an extremely large value for carnivores-flowers and vehicles-1-herbivores. This ordering is mildly inconsistent with the distances reported by Task2Vec in Fig. 3b the distance for vehicles-1-vehicles-2 is the highest here. Broadly, Task2Vec also results in a distance matrix that suggests that all tasks are equally far away from each other. As has been reported before (Li et al., 2020), this experiment also demonstrates the fragility of fine-tuning.

Recall that distances for uncoupled transfer in Fig. 3d can be comparable directly to those in Fig. 3a for coupled transfer. Task distances for the former are always larger. Further,

distance estimates of uncoupled transfer do not bear much resemblance with those of fine-tuning; see for example the distances for vehicles-2-carnivores, flowers-carnivores, and vehicles-1-vehicles-2. This demonstrates the utility of solving a coupled optimization problem in (17) which finds a shorter trajectory on the statistical manifold.

**Verifying the convergence of coupled transfer** We use an iterative algorithm to approximate the optimal couplings between source and target data. Fig. 4a shows the evolution of training and test loss as computed on samples of the interpolated distribution after 4 iterations of (17). As predicted by Thm. 4 the generalization gap. The training loss increases towards the middle; this is expected because the interpolated distribution is maximally far away from both the source and target data distributions at this point. The convex combination in (17d) keeps computations tractable but could also be a cause for this increase.

We typically require 4–5 iterations of (17) for the task distance to converge; this is shown in Fig. 4b for a few instances. This figure also indicates that computing the transport coupling in (11) independently of the weights and using this coupling to modify the weights, as done in say (Cui et al., 2018), results in a larger distance than if one were to optimize the couplings along with the weights. The coupled transfer finds shorter trajectories for weights and will potentially lead to better accuracies on target tasks in studies like (Cui et al., 2018) because it samples more source data.

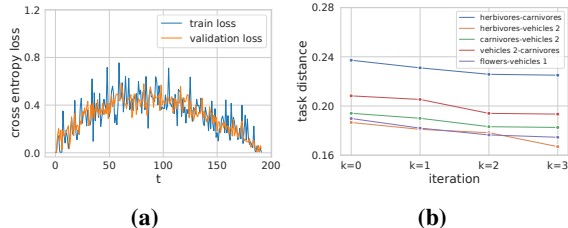

**(a)**        **(b)**

**Figure 4:** Fig. 4a shows the evolution of the training and test cross-entropy loss on the interpolated distribution as a function of the transfer steps in the final iteration of coupled transfer of vehicles-1-vehicles-2. As predicted by Thm. 4, generalization gap along the trajectory is small. Fig. 4b shows the convergence of the task distance with the number of iterations $k$ in (17); the distance typically converges in 4–5 iterations for these tasks.

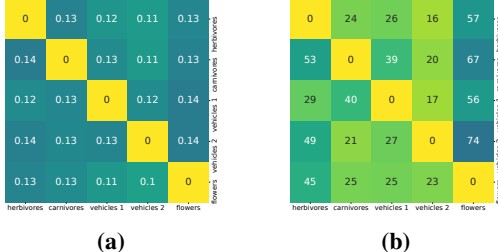

**(a)**        **(b)**

**Figure 5:** Fig. 5a shows the task distance using coupled transfer and Fig. 5b show the fine-tuning task distance. The numbers in Fig. 5a can be directly compared to those in Fig. 3a. The larger WRN-16-4 model predicts a smaller task distance for all pairs compared to the smaller convolutional network in Fig. 3a.

**Larger capacity results in smaller task distance**   We next show that using a model with higher capacity results in smaller distances between tasks. We consider a wide residual network (WRN-16-4) of Zagoruyko and Komodakis (2016) and compute distances on subsets of CIFAR-100 in Fig. 5. First note that task distances for coupled transfer in Fig. 5a are consistent with those for fine-tuning in Fig. 5b. Coupled transfer distances in  Fig. 5a are much smaller compared to those in Fig. 3a. This is consistent with our argument in Sec. 2.1. Roughly speaking, a high-capacity model can learn a rich set of features, some discriminative and others redundant not relevant to the source task. These redundant features are useful if target task is dissimilar to the source. This experiment also demonstrates that the information-geometric distance computed by coupled transfer can be compared directly across different architectures; this is not so for most methods in the literature to compute distances between tasks. This gives a constructive strategy for selecting architectures for transfer learning.

## 5   Related work

**Domain-specific methods**   A rich understanding of task distances has been developed in computer vision, e.g., Zamir et al. (2018) compute pairwise distances when different tasks such as classification, segmentation etc. are performed on the same input data. The goal of this work, and others such as (Cui et al., 2018), is to be able to decide which source data to pre-train to generalize well on a target task. Task distances have also been widely discussed in the multi-task learning (Caruana, 1997) and meta/continual-learning (Liu et al., 2019; Pentina and Lampert, 2014; Hsu et al., 2018). The natural language processing literature also prevents several methods to compute similarity between input data (Mikolov et al., 2013; Pennington et al., 2014).

Most of the above methods are based on evaluating the difficulty of fine-tuning, or computing the similarity in some embedding space. It is difficult to ascertain whether the distances obtained thereby are truly indicative of the difficulty of transfer; fine-tuning hyper-parameters often need to be carefully chosen (Li et al., 2020) and the embedding space is not unique. For instance, the uncoupled transfer process that modifies the input data distribution will lead to a different estimate of task distance.

**Information-theoretic approaches**   We build upon a line of work that combines generative models and discriminatory classifiers (see  (Jaakkola and Haussler, 1999; Perronnin et al., 2010) to name a few) to construct a notion of similarity between input data. Modern variants of this idea include Task2Vec (Achille et al., 2019a) which embeds the task using the diagonal of the FIM and computes distance between tasks using the cosine distance for this embedding. The main hurdle in Task2Vec and similar approaches is to design the architecture for computing FIM: a small model will indicate that tasks are far away. Achille et al. (2019b,c) use the KL divergence between the posterior weight distribution and a prior to quantify the complexity of a task; distance between tasks is defined to be the increase in complexity when the target task is added to the source task. This is an elegant formalism to define task distances but instantiating these ideas for deep networks requires drastic approximations, e.g., a Gaussian posterior on the weight space.

**Model complexity**   Learning theory typically studies out-of-sample performance on a single task.   Our goal is to account for the model complexity while defining the task distance. Complexity measures such as VC-dimension (Vapnik, 1998), come with a number of caveats when applied to deep networks because these measures are not reparameterization invariant. We exploited the geometric characterization of the statistical manifold Amari (2016) that leads to invariant quantities such as the Fisher-Rao distance.

**Coupled transfer of data and the model**   A key idea of our work is to observe that the marginal on the input can be transported in addition to the weights of the model. This is motivated from two recent studies. Gao and Chaudhari (2020) develop an algorithm that keeps the classification loss unchanged across transfer. Their method interpolates between the source and target data distribution using a mixture distribution (we use it as a baseline, see Sec. 3.1 and Sec. 4.3). Our work exploits this idea and computes the optimal way to modify both the input distribution and the weights. We use ideas from optimal transportation to compute the transport on input data; see Cui et al. (2018) who also solve an optimal transport problem approximately to estimate task distances. Coupled transport problems on the input data are also solved for unsupervised translation (Alvarez-Melis and Jaakkola, 2018).

## 6   Discussion

Our work is an attempt to theoretically understand when transfer is easy and when it is not. An often over-looked idea in large-scale transfer learning is that the dataset need not remain fixed to the target task during transfer. We heavily exploit this idea in the present paper and develop an optimization framework to adapt both the input data distribution and the weights from the source to the target. Although a metric is never unique, this gives legitimacy to our task distance. We compute the *shortest* distance in information space, i.e., the manifold of the conditional distributions. It is remarkable that this concept is closely related to the intuitive idea that a good transfer algorithm is one that keeps the generalization gap small during transfer, in particular at the end on the target task.

The most drastic approximation in this paper was to forgo a generative framework for the input distribution. This opens an interesting direction for future work which formulates the distance between tasks simply as the shortest geodesic on the manifold of joint distributions $p_w(x, y)$.

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
