# OpenReview forum: "An Information-Geometric Distance on the Space of Tasks"
_NeurIPS.cc/2020/Workshop/DL-IG — NeurIPSW 2020: DL-IG Oral_

### Official Review · AnonReviewer1 · 2020-10-22
**Review of "An Information-Geometric Distance on the Space of Tasks"**

**Rating:** 7
**Confidence:** 3

**Review:**

The explanation of what is happening in equations (6a)-(6d) is not clear. I think it will be useful to explain this in more detail since this is the main contribution of the paper. In particular, the weights are evolving on a path whose length is being calculated, but at the same time the distribution is also evolving from the source distribution to the target distribution. A slower-paced and in-depth explanation of what exactly is happening will add clarity to the paper. As of now, the paper has interesting ideas, but it is quite hard to decipher.

Minor comments

Line 52 should be p^s(y)
Define H in equation (4). Is entropic regularization necessary?
Line 75-77. Why is C calculated using a neural network? Isn't C_ij just the squared \ell_2 norm? Section 2.2 is a bit confusing. In equation (5), the interpolating path appears to go from an empirical distribution over (x,y) to another empirical distribution. So is the cost C_ij dependent on both x and y? Or is it just dependent on x? I assumed it depended only on x since line 69 states that only the empirical on x is being transported.
Eqn (6b) is missing a d\tau in the integral

---

### Official Review · AnonReviewer2 · 2020-11-07
**Review on "An Information-Geometric Distance on the Space of Tasks"**

**Rating:** 9
**Confidence:** 4

**Review:**

The work attempts to explore an interesting theoretical research avenue in transfer learning: in what set of tasks the transfer learning is easy or difficult. The paper begins with the observation that in large-scale transfer learning the dataset need not remain fixed to the target
task during transfer, and then develops an optimization framework exploiting this observation. The proposed framework is claimed to adapt both to the input data distribution and to the weights from the source to the target.
A more interesting part for me is a definition of distance in the manifold of conditional distributions and the conclusion that an adequate transfer learning algorithm is the one which keeps the generalization gap small during transfer.
Though it is an important work in the respective area, the authors may want to write it with a better logical flow. This is kind of a work that could be written without throwing too many technical jargon, rather by providing a lot of intuitions. With this minor suggestion, I recommend to accept this paper for the workshop.

---

### Decision · Program_Chairs · 2020-11-07

**Decision:**

Accept (Oral)

**Comment:**

I recommend this work for oral presentation. It would be nice to hear about it specifically the intuitive explanation of the results from either of the authors.